# Bioengineering of Soybean Oil and Its Impact on Agronomic Traits

**DOI:** 10.3390/ijms24032256

**Published:** 2023-01-23

**Authors:** Huan Song, David C. Taylor, Meng Zhang

**Affiliations:** 1Soybean Research Institute, Keshan Branch of Heilongjiang Academy of Agricultural Sciences, Keshan, Qiqihar 161606, China; 2College of Agronomy, Northwest A&F University, Yangling 712100, China

**Keywords:** soybean, bioengineering, fatty acid composition, seed oil content, agronomic trait

## Abstract

Soybean is a major oil crop and is also a dominant source of nutritional protein. The 20% seed oil content (SOC) of soybean is much lower than that in most oil crops and the fatty acid composition of its native oil cannot meet the specifications for some applications in the food and industrial sectors. Considerable effort has been expended on soybean bioengineering to tailor fatty acid profiles and improve SOC. Although significant advancements have been made, such as the creation of high-oleic acid soybean oil and high-SOC soybean, those genetic modifications have some negative impacts on soybean production, for instance, impaired germination or low protein content. In this review, we focus on recent advances in the bioengineering of soybean oil and its effects on agronomic traits.

## 1. Introduction

Soybean seed accumulates approximately 20% lipid and 40% protein and about 30% carbohydrate, making it an ideal feedstock in the food and feed industries. The high protein content of soybean economically compensates for its lower lipid content as an oil crop. Soybean not only meets the nutritional requirements of humans and animals but also plays important roles in chemical and health-related industries. Additionally, environmentally friendly symbiotic bacteroids endow soybean (as well as other leguminous plants) with abundant root nitrogen fixation capacity [1,2]. Therefore, policies and measures have been adopted around world to boost soybean production for a broad range of applications [3]. Global oil crop production, yields and growing areas increased by 240%, 48% and 82%, respectively, from 1983 to 2013 [4]. Next to oil palm, soybean production is the highest among the oilseed crops of the world, accounting for more than half of global production from 2012 to 2016 [4]. Considerable effort has been expended to diversify soybean oil for both food and industrial uses. Indeed, over the past forty years, tremendous progress has been made in modifying the oil of soybean and other oilseed crops to meet the needs of emerging bio-economies [1,2,5,6].

As one of the predominant edible oils, there are five common fatty acids in traditional soybean oil, including approximately 10% palmitic (16:0), 4% stearic (18:0), 20% oleic (18:1^Δ9^), 55% linoleic (18:2^Δ9,12^) and 10% α-linolenic (18:3^Δ9,12,15^) acids. Notably, its native fatty acid profile sometimes cannot meet nutritional or industrial specifications. It is worth noting that uncultivated plants are promising candidates as a substitute for oil crops to produce specific lipid components (e.g., high oleic acid, high omega-3 fatty acid). However, the adoption of these unconventional plants is not only limited by the plant oil composition but also by cropping systems, climate, yields and other restrictive factors. Therefore, genetic manipulation of specific characteristics and the high production of common oil crops have gained considerable interest, instead of exploiting other native species [7].

All five common fatty acids have been modified by genetic engineering in soybean oil to meet the expanding needs of end-users [1]. For example, a commercial soybean oil with high oleate levels may be superior to conventional oil in terms of oxidative stability [1]. Similarly, increasing the level of omega-3 linolenic and lowering the content of omega-6 linoleic in soybean meet the target for the prevention of obesity and cardiovascular diseases [8]. The demand of vegetable oils is growing rapidly, and the current annual consumption of 200 million metric tons may be doubled by 2050, which will impose a great burden on the limited availability of arable land [9]. Enhancing the oil content of crops, such as soybean, is of paramount importance for maintaining high yields without substantial impingements on non-crop land [6]. Improvement in soybean oil accumulation has drawn considerable attention because of its lower oil content compared to other oil crops like canola and sunflower.

The successful manipulation of FA composition and oil content largely depend on a deep understanding of lipid metabolic pathways and their regulatory networks [10,11]. At least two evolutionary polyploidization events in soybean have resulted in nearly 75% of genes having multiple copies, and the ensuing gene diversification and loss have remolded its genomic architecture [12]. Therefore, more research is required to understand the role of paralogs involved in lipid metabolism. Moreover, gene editing in soybean by simultaneously targeting multi-copy genes is sometimes more difficult than that in plants with a simple genome, such as *Arabidopsis* [13]. However, mutations in all multi-copy genes can be achieved through highly efficient CRISPR/Cas9 editing or RNA interference (RNAi) hairpin-targeting of high homology gene pairs [14,15]. Other methods, such as TILLING-by-Sequencing, can also be applied to multi-copy mutations accompanied by gene function analysis [16]. Here we have summarized some gene functions related to oil accumulation in soybean seeds (Figure 1 and Appendix A), a cornerstone for designing lipid improvement strategies.

Although tremendous progress has been made in soybean genetic improvement, modification of the fatty acid composition or lipid content sometimes has a detrimental effect on other agronomic traits, leading to declined yields or low-quality seed products [17,18]. During the improvement of seed oil, it is imperative to maintain the desirable agricultural traits of each soybean cultivar. Furthermore, promising phenotypes of genetically modified lines should have easily identifiable genetic markers and remain stable through multiple generations in multiple growing regions and cultivation years. Effective and new strategies for improving soybean oil are arising from advanced genome sequencing, multiple “omics” and highly efficient gene editing technologies [13,19,20]. Additionally, the detrimental impacts on other agronomic traits have been gradually overcome, although many underlying problems must be addressed in the future. Comprehensive reviews of soybean lipid modification have been published in the last two decades; these mainly focused on genome editing technologies, the effects of plant hormones and stress on oil content or tailoring fatty acid composition through the integration of conventional and modern breeding techniques [1,2,16,21]. However, there has been little focus on the impacts of those improvements on other agronomic performance factors. In this review, we highlight achievements made in the remodeling of fatty acid proportions and oil content in soybean seeds and focus on their impacts on agronomic traits. Studies in other plants have been referenced for inferring potential outcomes of lipid metabolic engineering if applied to soybean. Possible solutions to balance the improvement of seed oil and stability of agronomic traits are discussed.

## 2. Modifications of Soybean Oil Composition

Increasing omega-3 and reducing omega-6 FAs would bring soybean oil more in line with the recommended daily amount (RDA) ratio for human nutrition. On the other hand, oil with high levels of saturated fatty acids (SFAs) and mono-unsaturated FAs (MUFAs) show excellent oxidative stability and a high melting point, which endow superior performance for baking applications. Soybean oil with low polyunsaturated fatty acids (PUFAs) mitigates the need for extensive hydrogenation, a process that results in significant proportions of *trans*-fatty acids, which can negatively affect cardiac health [47]. Accumulating a certain fatty acid may meet a specific demand in a defined application. Here, we will update the review by Clemente and Cahoon (2009) by discussing more recent attempts to improve soy oil quality while reducing impacts on other agronomic traits.

### 2.1. High-Oleic Acid Soybean Oil

The desaturation of stearic acid (18:0) to oleic acid (18:1) is catalyzed by stearoyl-acyl carrier protein desaturase (SAD) in plastids. After the removal of acyl carrier protein (ACP) by acyl-ACP thioesterase, the fatty acids are then activated to acyl-CoAs by a long-chain acyl-CoA synthetase (LACS) before being transported to the endoplasmic reticulum (ER) for assembly into glycerolipids. The resulting 18:1-CoA is transferred to phosphatidylcholine (PC) where it is further desaturated to linoleic and linolenic acids by the fatty acid desaturases (FADs) FAD2 and FAD3, respectively, in the ER [48]. Most of the research related to high-oleic acid crops has focused on preventing or mitigating flux from 18:1 to 18:2, controlled by the *FAD2* family [14,30,31]. There are seven *FAD2* homologs in the soybean genome, which diverged into two groups in the phylogenetic tree, including two *GmFAD2-1*s and five *GmFAD2-2*s. Both *GmFAD2-1*s are highly expressed in seeds, while *GmFAD*2-2 members exhibit different expression patterns (except for the non-expressed *GmFAD2-2A* with 100 bp truncated in its CDS) [30]. Simultaneously knocking-out two *GmFAD2-1*s resulted in 80% oleic acid accumulation at the expense of linoleic acid (reduced to 1.3~1.7%) without compromising the protein and oil content in seeds [14]. Single mutants of *GmFAD2-1a* and *1b*, generated by TILLING, show 127% and 47.2% increases in oleic acid compared with WT levels, respectively [31]. Similarly, the oleic acid content in the *fad2-1a/1b* double mutant obtained via transcription activator-like effector nucleases (TALENs) also rose to 80%, while linoleic acid was reduced to 4% [49]. Mutations in two *FAD2-1*s and *FAD3A* further increased the 18:1 level to 83.2–85.9% [50,51]. These results suggest that *GmFAD2-1*s, rather than *GmFAD2-2*s, play the dominant role in flux from 18:1 to 18:2 in seeds. Surprisingly, some mutants created using Cas9 and TILLING-by-Sequencing+ technology showed that *GmFAD2-2*s also make significant contributions to the flux from 18:1 to 18:2 in seeds [30,52]. Knocking-out *GmFAD2-1A* and *GmFAD2-2A* markedly increased oleic acid from 19.15% to 72.02% in T_3_ lines with a concomitant improvement in protein, while seeds of these lines were smaller in size with a deeper colored seed coat than the wild-type [53]. Considering the high sequence similarity among *GmFAD2s*, [30,52] one cannot exclude the possibility of off-target genome editing. It is also unknown whether there are interactions among these GmFAD2s. Cooperative relationships and post-transcriptional modifications, such as phosphorylation, in FAD2 members remain to be deciphered in the future [54]. Notably, PUFAs can be oxidized by lipoxygenases (LOXs) to produce hydroperoxides, which are further degraded into hexanal, 3-cis-hexenal, etc. This enzymatic degradation of PUFAs is a main cause of the off-flavor in soybean [55]. It will be interesting to evaluate the change in the beany flavor of high-oleic acid soybean, in which PUFAs are largely decreased.

Crop germplasms with significantly increased oleic acid content may have detrimental effects on agronomic performance (e.g., plant height, seed germination), which ultimately leads to low quality or yields [56,57]. For example, suppression of *FAD2* in rice largely affects starch properties; a higher temperature is required for dissociating the more stable amylose–lipid complex in cultivars enriched in 18:1 compared to the temperature required for those high in 18:2 [58]. A mutant line of rapeseed with an oleic acid content above 80% showed retardation in plant morphogenesis and 7–11% reductions in seed oil content, but an increase in protein content [59]. Silencing *GhFAD2–3* via RNAi in cotton severely arrested anther development, which illustrates that the biosynthesis of 18:2 is essential for male fertility [60]. There have been no reports of adverse effects on normal physiology in high 18:1 soybean germplasms; nevertheless, there is ample evidence that *FAD2*s are indispensable for stress resistance both in plants and other eucaryotes [61,62,63]. In soybean, the germination rate of high-oleic acid G00-3213 near-isogenic lines (NILs) was significantly lower than that of normal NILs when planted in cold soil [64]. Moreover, reducing polyunsaturated fatty acid content in vacuolar and plasma membranes in *fad2* is associated with a lower Na^+/^H^+^ exchange activity in *Arabidopsis*, which makes it more susceptible to salt stress [65]. Predictably, vacuole-localized GmFAD2-2B may be involved in salt tolerance, which should be taken into consideration during gene manipulation [30]. Collectively, the data suggest that soybean germplasms with increased 18:1 contents in oil should be further evaluated for ancillary impacts on plant growth and development, especially under stress conditions. Fortunately, the differential expression in various tissues and subcellular localization of GmFAD2s provide the possibility of manipulating specific *FAD2*s without detrimental effects on agronomic traits [30]. For example, down-regulating a seed-specific *FAD2* or retaining one intact *FAD2* homolog could be alternative strategies for retaining high levels of 18:1 in seeds without any hindrance to plant growth [57,66,67]. Attenuating FAD2 activity via C-to-G base editing or by reducing its expression level through high frequency targeted edits in *cis*-regulatory elements of the *FAD2* promoter provide new paths for accumulating oleic acid with less adverse effects on growth [68,69]. Therefore, the seed-specific knockout or silencing of *GmFAD2* paralogs may avoid negative impacts on soybean growth and development.

### 2.2. High-Linolenic Acid Soybean Oil

The conversion of 18:2 to 18:3 is catalyzed by omega-3 fatty acid desaturase (FAD3) in the ER [48]. 18:3 is essential for the prevention of cardiovascular and cerebrovascular diseases in humans and animals [8]. In the soybean genome, four genes encoding FAD3 desaturases are designated as *GmFAD3A* (*GmFAD3-1b*), *GmFAD3B* (*GmFAD3-1a*), *GmFAD3C* (*GmFAD3-2a*) and *GmFAD3E* [32,70]. Transcripts of *FAD3A*, *FAD3B* and *FAD3C* exhibit high expression levels in developing seeds and stems but lesser expression in root nodules and pods and are only weakly expressed in green leaves [70], while *FAD3E* is mainly expressed in the root tip and moderately expressed in developing seeds (https://www.soybase.org; accessed on 8 March 2022). In soybean, *FAD3A* and *FAD3E* may be major contributors to the linolenic acid content in seeds compared with the other two homologs [32,70], and in rice transformants, FAD3B and FAD3C may have much lower activities for 18:2 desaturation [33,71]. The linolenic acid content in soybean seeds is relatively low (8%), which may be due to weak desaturase activity or the post-translational regulation of native FAD3 [71,72,73]. Despite 18:3 improvements mediated by the over-expression of *GmFAD3A* in rice, endogenous *OsFAD3* manipulation proved to be more efficient than exogenous *GmFAD3A* for rice improvement [71]. In addition to the low desaturase activity in soybean, phosphatidylcholine: diacylglycerol acyltransferase (PDAT) prefers to transfer 18:2 from PC to DAG to produce TAG [72]. A higher proportion of 18:3 (about 15%) remains in PC compared to that deposited in TAGs (about 3%) upon seed maturation [72].

To optimize the omega-6/omega-3 ratio in soybean or other oil crops, the introduction of exogenous *FAD3*s from high-18:3-TAG plants, such as *Physaria fendleri* or peony trees, provides approaches to substantially increase linolenic acid proportions [74,75]. Seed-specific expression of *PfFAD3-1* (driven by the promoter of soybean β-conglycinin) resulted in 42% linolenic acid content in soybean seeds, while only about 10% accumulated using a 35S constitutive promoter construct [8]. *PfFAD3* lines showed a significant improvement in seed yield and seed size, probably as a consequence of increasing adaptability to stress tolerance [8]. Surprisingly, some *FAD3* transgenic lines exhibited a decline in 18:3 proportions compared with that in untranformed controls, which may be attributed to co-suppression phenotypes, sometimes observed in *FAD2-*over-expression lines. The expression of *FAD3* derived from *F. moniliforme* in soybean led to a 6.5-fold increase in the 18:3 content (from 10.9% to 70.9%) without compromising the morphology and germination of seeds [76]. Via the ectopic expression of *GmFAD3A* in rice, oil content and seed germination rates under cold conditions were improved considerably [33].

As mentioned above, PUFA can be degraded by LOXs. Off-flavors may be increased in high-18:3 soybean, and lowering LOX activity should be considered in this case. A soybean mutant (*Gmlox1Gmlox2Gmlox3*) exhibiting the loss of lipoxygenase activity showed no impact on lipid and protein content [39]. However, since the LOX pathway is involved in germination and seedling growth, potential effects on physiological processes and anomalous growth phenotypes of lipoxygenase-free mutants should be carefully assessed [77]. Since *GmLOX2* and *GmLOX3* are not expressed in seeds (Appendix A), knocking out the seed-specific *GmLOX1* may be an alternative to balance the elimination of beany flavor and minimize its impact on plant growth.

However, in some species, such as *Arabidopsis*, high 18:3 levels in seeds affect the oil content because it strongly attenuates both embryo development and lipid accumulation [78]. A high accumulation of free 18:3 causes severe oxidative stress and ER anomalies in *CsFAD3-OE* developmentally defective embryos, which can be rescued by the co-expression of *CsLPAT*, which can incorporate the excess 18:3 into phosphatidic acid [78]. Phosphatidylcholine:diacylglycerol cholinephosphotransferase (PDCT) catalyzes interconversion between PC and diacylglycerol and enriches PUFAs in TAG [79]. Conversion from PC to DAG varies considerably among different species, even though they share close evolutionary relationships. For example, PDCT in *Arabidopsis* transfers 40% of 18:1 from PC to DAG, while in canola, only 18.2% of 18:1 can flux through PC for remodeling via PDCT [79,80]. PDCTs with high substrate affinity for C_18_-PUFA-PCs, have been isolated and expressed in attempts to engineer high PUFA-TAGs in crops. For example, the seed-specific expression of flax *LuPDCT1* and *LuPDCT2* in *Arabidopsis* resulted in 16.4% and 19.7% increases in C_18_-PUFAs, respectively [81]. Accordingly, *GmPDCTs* may be candidates for 18:3 improvement in soybean in the future.

### 2.3. Low-Linolenic Acid Soybean Oil

Three double bonds of 18:3 make it easier to be oxidized than other usual fatty acid. Oxidation rates of 18:3 are twice that of 18:2 [82]. Therefore, to extend the shelf life, some researchers have targeted *FAD3* reduction to control α-linolenic acid levels in soybean seeds. The triple-mutant ‘LOLL-A9’ developed using a soybean TILLING system, produces <2% 18:3 in seeds [83]. Ultra-low 18:3 with 1% content has been obtained via the seed-specific silencing of all *FAD3*s without any perturbation to seed germination and yields [84].

However, varying results have been reported on the impacts of silencing *GmFAD3*. Silencing of *GmFAD3E* led to a lower oil seed content (approximately 1% reduction), which may be due to the up-regulation of a negative regulator (biotin attachment domain containing proteins, BADCs) of ACCase [32]. Seed protein content was elevated and an allergen (C6T3L5) was reduced [32]. BPMV-mediated *GmFAD3*-silenced plants showed negative effects in leaf morphology, but exhibited desirable agronomic traits, such as high seed yields, without compromising the protein and lipid content [85]. Polyunsaturated fatty acids are vital constituents of cell membranes and therefore key determinants of membrane fluidity in plants [48,86]. Additionally, free or esterified linolenic acid is a precursor for the synthesis of various biomolecules, such as jasmonate, which mediates biotic stress responses [87]. Rice exposed to cold treatment is vulnerable to damage from reactive oxygen species (ROS), leading to lower fluidity of the thylakoid membranes and photosystem I inactivation [86]. The expression of *GmFAD3A* in rice significantly improved cold acclimation both in the budding seed and in young seedlings [33]. Moreover, the over-expression of *GmFAD3A* in soybean enhanced drought and salinity stress tolerances, whereas it weakened heat stress performance [88]. In contrast, soybean plants with *GmFAD3A* silenced using a BPMV vector were more vulnerable to drought and salinity stress, but had greater endurance under heat stress [88]. Notably, *GmFAD3*-silenced plants are susceptible to the virus *Pseudomonas syringae*, but are resistant to the oomycete pathogen *Phytophthora sojae,* which may be due to the accumulation of salicylic acid (SA) [85].

In summary, silencing *GmFAD3s* to lower the linolenic acid content may have deleterious effects on physiological processes, such as photosynthesis, transpiration and phytohormone signaling. As in the case of *FAD2* bioengineering, one must beware of the coincident risks of down-regulating *FAD3* in vegetative tissues. Since three of the four *GmFAD3*s are highly expressed in seeds, seed-specific silencing may be an alternative strategy to retain essential FAD3 activity in non-reproductive tissues.

### 2.4. Low-Saturated FA Soybean Oil

Despite many potential applications in industry, saturated fatty acids (SFAs) have adverse effects on human health; accordingly, progress has been made to create low-SFA soybean oil. In plants, FAs are synthesized de novo in plastids [89]. Acyl-ACP thioesterases (FATs) in different species determine the chain length of nascent fatty acids leaving the plastid for incorporation into oils [89]. These thioesterases are often critical targets for lipid profile modification. In general, the gene family of *FAT*s can be classified into two clades, namely *FATA* and *FATB*, for hydrolyzing 18:1-ACP and other acyl-ACPs, respectively. In total, 12 genes encoding acyl-ACP thioesterases have been identified in soybean [34]. Only two of them are *FATA*s; the other ten, *GmFATB*s, are classified into three groups. Tissue profiling shows that two *GmFATA*s, *GmFATB1A/B* and *GmFATB2A* exhibit relatively high expression in developing seeds [34]. Total saturated fatty acid content decreases drastically in leaves and the seeds of single mutants of *fatb1a* and *fatb1b*, and double mutant *fatb1a/1b* plants exhibit male sterility, a dwarf phenotype and other undesirable traits [35]. Many studies have emphasized the importance of *FATB* in wax biosynthesis, plant growth and lipid accumulation [90]. The manipulation of *FATA/FATB* sometimes causes retardation in acyl flux toward lipid synthesis and impairs seed germination [91,92]. Pollen development n *fatb2a/2b* mutants should be evaluated because both genes are intensely expressed in flowers [34]. A *fata1a*-mutant created via TILLING-by-sequencing shows a decrease of 16:0 from 11.6% to 9.1–10.5%, as well as an increase in oleic acid [34].

Acyl carrier protein (ACP) in the plastid is an essential cofactor for de novo FA biosynthesis by the FA synthase complex [24]. The *GmACP*s and *ACP*s of other legume species can be clustered into a legume-specific subclade, suggesting that the ACPs that they encode may share a similar function [24]. In fact, *GmACP* function is related to numbers of root nodules. The suppression of *GmACP* led to a significant reduction in SFAs in roots and a concomitant decline in the nodule number [24]. However, whether the manipulation of ACP levels can be used for seed SFA improvement requires further investigation.

The conversion of stearic acid to oleic acid is catalyzed by stearoyl-ACP desaturase (SAD or SACPD) in the plastid, making it an appropriate target for controlling SFA content in crops [93]. Five *SACPD*s have been characterized in the soybean genome. *SACPD-A* and *SACPD-B* are ubiquitously expressed, while *SACPD-C* is highly expressed in seeds, roots and nodules. *GmSACPD-D* is expressed in most tissues, and *GmSACPD-E* might be a pseudogene [44]. However, recent work has indicated that the over-expression of *SACPDs* is not an appropriate means to improve oleic acid in some crops, but an effective way for PUFA accumulation and the reduction of saturated fatty acids. For instance, in corn and *Arabidopsis*, the seed-specific over-expression of *ZmSAD* contributed to a lower ratio of saturated to unsaturated fatty acids but no change in the oleic acid content [94]. The over-expression of *SsSAD* from *Sapium sebiferum* (L.) Roxb in *Brassica napus* resulted in an obvious increase in the PUFA content along with a substantial decline in 18:1 at cold temperatures, thereby improving the freezing tolerance of canola [95]. Similar results in freezing tolerance and cold acclimation have been obtained in experiments in potato [96]. Therefore, the function of *GmSACPD*s in seed fatty acid composition in soybean should be evaluated in the future.

### 2.5. High-Saturated FA Soybean Oil

Stearic acid content in common soybean oil is below that recommended for baking applications (20%) [47]. *FAT*s are promising candidates for manipulation in order to elevate saturated fatty acids in crop oils. The over-expressing *GmFATB1* in soybean only slightly increased the palmitic content and had no effect on the stearic acid content, suggesting that soybean has evolved a counterbalance mechanism to avoid the over-accumulation of SFAs [35]. In contrast to that in soybean, seed specific over-expression of *FATB1* in *Arabidopsis* significantly increased the proportion of 16:0 with normal plant growth [97]. Soybean lines with dual-silenced endogenous *FATB* and *FAD2* and the over-expression of *FATB* from mangosteen produced 20% 18:0 in seeds without yield penalties, providing a soybean oil that is more suitable for baking applications [47]. Nevertheless, there is a negative correlation between the SFA relative content and total FA content, which must be taken into consideration. For example, in *FAT* transgenic *Arabidopsis* and *B. juncea*, the saturated fatty acid content was markedly improved but with a strong reduction in the oil content [98,99]. Additionally, seed morphology and the germination of some transgenic lines were severely impacted in transgenic *B. juncea* over-expressing the *Madhuca longifolia MlFatB* [98].

Silencing *SACPDs* is another approach for improving SFAs in soybean oil (Figure 2). A homozygous double mutant of *GmSACPD-B* and *GmSACPD-C* shows an increase in stearic acid content from 4.3% to 14.6% and a 1.2% decrease in total oil [100]. Stearic acid is elevated to 13.5% as a result of mutations in *SACPD-C* [93]. As a nodule-expressed gene, *SACPD-C* is of paramount importance for nodule morphology, physiology and biotic defense responses [101,102]. Furthermore, nodules of the s*acpd-c* mutant showed low expression of *Nitrogenase D* and premature senescence with central cavities [101,102]. Whether down-regulated *SACPD-C* has profound effects on plant yields requires further investigation. Fortunately, mutations in non-conserved residues of SACPD-C provide a target for increasing stearic acid while maintaining healthy nodules [16,45]. Unlike that with *GmACP*s, *sacpd-c* mutant lines show no obvious increase of stearic acid content in roots, and the stearic acid content in seeds of lines produced by substitutions in the conserved residues is much higher (up to 20.11%) than that in lines created with substitutions in non-conserved residues (stearate levels of up to 7.24%) [45]. Aside from affecting nodule development, *SACPD-C* plays a role in leaf structure and morphology [45]. The *sacpd-c* mutant exhibits undulated leaves due to leaf cell disorganization, which may result from higher FA saturation levels in membranes of leaf tissue compared with those in the WT [45]. Moreover, knocking down *SACPD-A/B/C* restricts plant growth and causes female sterility in tobacco [103].

*SACPD*s are also involved in plant pathogen defense. For instance, *SA INSENSITIVITY OF npr1-5* (*SSI2*) encodes a stearoyl-ACP desaturase (SAD) that has been reported to simultaneously enhance resistance and repress growth. In *Arabidopsis* an *ssi2-2* mutant accumulates more than twice the proportion of SA compared with that in the WT, and genes of the pathogenesis-related SA pathway are up-regulated in *ssi2-2* which is not susceptible to the bacterial pathogen *Pst DC3000* [104]. As previously reported, silencing *SACPDs* conferred soybean resistance against some bacterial pathogens and impacted agricultural traits like seed size, morphology and plant height [105]. Collectively, these results indicate that defense signaling pathways mediated by altering the oleic acid content are relatively conserved among different plant species. In conclusion, SFAs play multiple important roles in plant development, including stress adaptation, plant–microbial interactions and phytohormone responses, and fully understanding these impacts is critical for seed oil modification. Notably, sub-functionalization of these genes should be evaluated to avoid pleiotropic effects on soybean agronomic traits.

### 2.6. Production of Unusual FAs (UFAs) in Soybean Oil

UFAs, with structural variations in the chain length, branches, number, position and configuration of unsaturation, etc., exhibit valuable properties that may be applied in chemical industries. To date, nearly 500 UFAs have been identified in vascular plants [108]. Understanding the mechanisms of UFA biosynthesis provide knowledge for engineering their production in oil crops. Here, we summarize advancements in the production of three UFAs in soybean.

#### 2.6.1. Epoxy Fatty Acids (EFAs)

Vernolic acid (VA), a monounsaturated epoxy fatty acid, can be used as a renewable chemical feedstock. Its derivatives have been explored for many industrial applications, such as surface coatings, glues and plasticizer substitutes. Epoxygenase (EPX), a member of the *FAD2* super-family, catalyzes the formation of vernolic acid from linoleoyl moieties esterified to PC, a substrate that is in rich supply in soybean seeds. However, the seed-specific expression of *SlEPX* from *Stokesia laevis* in soybean resulted in only 7% VA production with negative impacts on seed morphology, oil and protein contents [18,106]. It was reported that the accumulation of hydroxy fatty acids in membrane lipids, such as PC, may be simultaneously detrimental to plant growth and TAG formation [109]. The co-expression of *VgDGATs* from *Vernonia galamensis* with *SlEPX* in soybean reduced the level of VA in membrane lipids and increased its accumulation in TAGs up to 27% [18]. More importantly, this co-expression compensated for the reduction in oil and protein and transgenic seeds reverted to a normal size and morphology [18]_._ Accordingly, exploiting the use of DGATs or PDATs with preferences for VA should be helpful in improving the transfer of VA from membrane lipids to TAGs.

#### 2.6.2. Conjugated Fatty Acids

Non-methylene-interrupted double bonds distinguish conjugated fatty acids from common polyunsaturated fatty acids. Conjugated fatty acids (CFAs) have many potential applications in industry because they are easily oxidized [110]. Additionally, they have excellent potential as bioactive molecules, can promote balanced digestion and are anti-carcinogens in human and animal health [111]. Fatty acid conjugases (FADXs), also divergent forms of FAD2, catalyze linoleoyl moieties on PC to form Δ^11^ and Δ^13^ double bonds. However, it was found that bioengineering CFA accumulation in soybean resulted in problems similar to those encountered with VA accumulation, such as wrinkled and low-vigor seeds [110]. The over-expression of *FADX* from *Calendula officinalis* and *Momordica charantia* in soybean and *Arabidopsis* results in nearly 25% CFAs in PC from seeds, but the CFAs cannot be as efficiently transferred to TAG as in their native plants [110]. Because *Arabidopsis* and soybean lack the necessary detoxification pathways, more than 23% of the CFAs remained in PC and were not transferred to TAG. In contrast, plants enriched with these unusual FAs, such as *C. officinalis*, can effectively exclude them from PC throughout seed development, thereby maintaining low-levels CFAs in membrane lipids [110]. Therefore, when engineering CFA biosynthesis, a highly efficient means of removing CFAs from PC for deposition in TAGs is essential for avoiding the negative effects on plant growth and agronomic performance.

Eleostearic acid (ESA) is a useful conjugated fatty acid for human health and industrial applications. The co-expression of tung *FADX* and *DGAT2* in *Arabidopsis* resulted in the exclusion of eleostearic acid from leaf phospholipids and elevated its accumulation in leaf oil, while developmental retardation was rescued and healthy seeds were produced [111]. Higher amounts of neutral lipid-containing ESA in leaves have been obtained by impeding TAG hydrolysis, which has almost a negligible impact on seed morphology and the germination rate [111].

Collectively, these successful precedents provide an important reference for soybean oil modification with higher conjugated fatty acids.

#### 2.6.3. Acetyl-TAGs

*Euonymus alatus* accumulates more than 90% acetyl-TAG in whole seeds, which is of high commercial interest [112]. Long-chain FAs have been replaced by acetate at the *sn*-3 position in acetyl-TAG, which confers a lower viscosity, freezing point and caloric value compared to those of common TAGs [112]. The heterologous expression of *EaDGAT1* or *EaDGAT2* resulted in the inability to produce and accumulate acetylated triacylglycerols in host yeast; however, a new member of the MBOAT gene family, *E. alatus* diacylglycerol acetyltransferase (*EaDAcT*), was isolated by conducting an in-depth investigation of transcriptomes within multiple tissues [112]. In contrast to EaDGATs, this acyltransferase accepts acetyl-CoA, but is incompatible with long-chain acyl-CoAs in vitro [112]. The expression of *EaDAcT* in camelina and soybean produced acetyl-TAGs comprising up to 70 mol% of the seed oil with normal seed germination and plant development [113]. Euonymus species from the Celastraceae family, such as *E. fortune* and *E. kiautschovicus*, also produce high seed levels of acetyl-TAGs. As an alternative, the functional investigation of EfDAcT and EkDAcT may further expand the capacity to produce acetyl-TAGs in soybean because they have higher activity toward acetyl-CoA than homologs like EaDAcT, in vitro [114]. Simultaneously suppressing endogenous *DGATs* further increased acetyl-TAG accumulation up to 80 mol%, but led to a 7.5–11% reduction in the total oil content in field-grown camelina [113]. Accordingly, it is promising to consider improving acetyl-TAG content in soybean by knocking down *GmDGAT*s in the transgenic host; however, a potential concomitant reduction in oil needs to be considered.

## 3. High-Oil Content Soybean

Eight reproductive (R) stages are used to describe soybean development. Stage R5 is characterized by rapid seed growth or seed filling, while stage R8 designates the mature seed (https://extension.umn.edu/). The rapid accumulation of lipid and protein simultaneously occurs at stages R6~R7 when starch accumulation begins to wane [115]. Given the negative correlation between lipid and protein accumulation in soybean seeds, it may not be meaningful to alter carbon flux toward one or the other, unless sufficient additional carbon is available. On the other hand, a fraction of lipid is degraded for raffinose family oligosaccharide (RFO) production during seed maturation [115]. Therefore, high-oil soybean may be achieved by reducing competing biosynthetic processes, thereby providing sufficient lipid precursors, improving the efficiency of oil biosynthesis and blocking lipid degradation (Figure 3).

### 3.1. Reduction in the Synthesis of Competitive Compounds or Inhibitors of Oil Accumulation

Recent genome-wide association (GWAS) analysis revealed that *Protein Oil Weight Regulator 1* (*POWR1*), a CCT-domain gene, showed pleiotropic effects on both the protein and lipid content [119]. A soybean accession with a 321 bp insertion of a transposable element (TE) in *POWR1* produced a truncated CCT domain and exhibited a significant increase (nearly 10%) in both the lipid content and 100-seed weight, but a reduction in the protein content of 5.3–7.1% [119]. Above all, this line produced 150.3 kg/ha higher yield than the non-insertion line. In contrast, lines without a TE insertion allele produce significantly higher protein (2.5%) together with a 2.4% reduction in the oil content and a 3.57 g decline in the 100-seed weight compared with those in non-transgenic control seeds. During soybean domestication, selection for *POWR1* conferred the simultaneous optimization of lipid content and seed yields. The reduction in protein content was compensated by about a 5% yield increase [119].

Given the fact that higher lipid content is often associated with lower protein levels, can a high-oil crop can be achieved by redirecting the substrate from protein to lipid synthesis? Considerable effort has been expended on this strategy in some species, but with limited success, since silencing the intrinsic major storage protein genes often leads to compensatory increases in other seed storage proteins without activating lipid accumulation [128]. Moreover, in rapeseed, the disturbance of napin and cruciferin deposition in seeds led to 10–15% less protein and lipid than those in the WT [129]. This common phenomenon is known as proteome remodeling. A successful means of lipid enhancement in *Arabidopsis* seeds was obtained through extension of the expression period for WRINKLED1 (WRI1; a master transcription factor regulating seed oil biosynthesis) in seed storage protein-knockout mutants [130]. In the future, the reciprocal relationship between oil and protein accumulation during seed development could be exploited in different species. Seed coat protein Bloom 1 (B1), enriched with allergens in *Glycine soja*, confers resistance to some predators but also precludes their use for human consumption [116]. Nevertheless, phenotypes with ‘no Bloom’ have gradually replaced B1 during artificial selection. The non-synonymous mutation (C to T) of *B1* may damage the helix structure of the protein [116]. Coincidently, because B1 acts as repressor of many key transcription factors involved in oil content, such as WRI1, ABI3 (ABSCISIC ACID INSENSITIVE 3) and LEC1 (Leafy cotyledon 1), *b1* mutant lines coordinately induced lipid accumulation in soybean [116]. In contrast, oil content was decreased in seeds of *B1* over-expression lines. Interestingly, the reduction in fatty acid biosynthesis mediated by *B1* is restricted to the pod coat and has no effect in other seed tissues [116]. The elevation of oil content mediated by manipulating *b1*-mediated pleiotropic effects has greatly expanded the scope of lipid improvement and seed dust elimination in soybean.

### 3.2. Enhancing Seed Oil Content by Providing a Sufficient Substrate

A weighted correlation network analysis (WGCNA) indicated that genes encoding enzymes in the glycolysis pathway, such as fructose-bisphosphate aldolase and NADH-glutamate synthase, are positively associated with oil accumulation during soybean seed development [131]. Glycerol-3-phosphate (G-3-P) and acetyl-CoA are produced from glycolysis, suggesting that providing a sufficient substrate for fatty acid biosynthesis and boosting the availability of G-3-P may augment carbon flux toward TAG. Recent work has demonstrated that the *Seed Thickness 1* (*ST1*) gene, encoding UDP-glucose 4-epimerase, involved in pectin biosynthesis, is an alternative target for lipid content improvement [121]. A NAD-binding domain embedded in ST1 may imply that it has a profound impact on lipid biosynthesis substrates, such as glycerol-3-phosphate [121]. Further experiments revealed that the over-expression of *ST1* enhances lipid accumulation by significantly increasing the expression of genes associated with glycolysis [121]. However, seed shape is widely different between *ST1* transgenic and wild-type lines [121]. During soybean domestication for yellow seed color, *ST1* was co-selected, making soybean seeds rounder, with a higher oil content [121].

As a primary precursor of acetyl-CoA for fatty acid biosynthesis, sucrose can be transported from photosynthesis tissues to storage tissues, mediated by both sucrose transporter (SUT) and sucrose effluxer (SWEET) [123]. Considerable progress has been made toward identifying members of the soybean *SWEET* family; among them, *GmSWEET10a* (also named *GmSWEET39*, *Glyma.15g049200*) shows a critical role in seed oil accumulation [122,123,124]. It is located in an approximately 40 kb genomic region, which has been identified as a selective sweep region, a major QTL region that simultaneously controls seed protein and lipid content, as well as seed size [124]. A haplotype of truncated GmSWEET39 with a C-terminus deletion (CC- allele) exhibits a tendency for oil improvement in soybean modern breeding [124]. However, the haplotype variation in *SWEET39* cannot explain the marked variability in lipid and protein content between *G. soja* and *G. max*, which may be attributed to additional major QTLs linked to seed protein and oil content [124]. Accordantly, haplotype analysis of *GmSWEET10b*, a close homologue of *GmSWEET10a*, showed dramatic differences in lipid and protein content between cultivated and wild soybean ecotypes, suggesting the ongoing selection of *GmSWEET10b* in modern soybean oil improvement programs [122]. Additionally, a recent study revealed that *GmST05* (Seed Thickness on Chromosome 5, a phosphatidylethanolamine-binding protein (PEBP) family member) plays roles in seed shape, seed protein and lipid content also affects the plant height and pod number per plant, which may act by regulating the transcription of *GmSWEET10a* [120]. Two *GmST05* haplotypes (*GmST05^HapI^* and *GmST05^HapII^*, natural variations in the promoter) display geographic differentiation in landraces and cultivars. The homozygous over-expression of *GmST05^HapI^* resulted in a 5% greater 100-seed weight, parallel with higher oil and lower protein contents [120]. Conversely, RNAi and *GmST05^HapII^* transgenic lines exhibited significantly lower oil but higher protein content than *GmST05^HapI^* lines [120]. Together, these studies suggest that improving substrate abundance for fatty acid biosynthesis is an effective way for enhancing soybean oil content with impacts on seed size and protein content.

### 3.3. Maximizing TAG Content by Boosting TAG Synthesis-Related Pathways

Activated acyl groups are sequentially esterified to TAG (Kennedy pathway) by acyl-CoA: glycerol-3-phosphate acyltransferase (GPAT), acyl-CoA: lysophosphatidic acid acyltransferase (LPAAT) and diacylglycerol acyltransferase (DGAT), respectively [132,133,134]. Through functional complementation assays using the yeast strain ZAFU1, a large number of *GmGPAT* genes was comprehensively characterized in soybean [36]. However, only GmGPAT9-2 exhibited high acyltransferase activity in serial dilution assays. Therefore, *GmGPAT9-2* was originally considered a candidate for lipid content improvement. Unexpectedly, the seed-specific expression of *GmGPAT9-2* failed to increase TAG accumulation in *Arabidopsis*, but resulted in accumulation of the very long-chain fatty acids 20:0 and 22:1 [36]. The utility of *GmGPAT* over-expression for improving oil accumulation should be further investigated in soybean. Additionally, two *GmLPAATs*, namely *Gm02LPAAT* and *Gm10LPAAT,* have been shown to enhance docosahexaenoic acid (DHA) at the *sn*-2 position in TAG in *Arabidopsis* seeds co-expressing DHA genes [40]. However, to date, similar experiments have not been conducted in soybean.

DGAT is a known rate-limiting enzyme for TAG synthesis in the Kennedy pathway [135,136]. In total, 10 *DGAT* family members in the soybean genome were divided into three groups, including three *DGAT1*s, five *DAGT2*s and two *DGAT3*s [26]. Simultaneously knocking down all three *DGAT1*s in soybean decreased oil accumulation while improving protein content and 100-seed weight and also prolonged the process of leaf senescence [27]. In another study, the seed-specific over-expression of a soybean *GmDGAT2A* increased oil content and linoleic acid and also promoted radicle elongation during germination without affecting other agronomic traits, such as plant height or pod number [28]. Moreover, in yeast expression studies, a variant with a truncation of 16 amino acids in the *N*-terminus intrinsically-disordered region (IDR) of GmDGAT2A shows further improvements in TAG production beyond that of lines containing transgenic full-length *GmDGAT2A* [28]. Recently, functional assays of *GmDGAT3-2* in yeast and tobacco have suggested that soybean with high oleic acid and oil content may be obtained via its over-expression [29]. DGAT1 homologs with high activity from other plants can also be used to increase oil content in soybean. Soybean lines expressing *VgDGAT1A* show increased seed oil by 4% without reductions in protein content or yield on a per-land-unit basis, compared with those in the wild-type [107].

Alternatively, PC can be an acyl donor to a DAG acceptor, mediated by PDAT [135]. *GmPDAT*, a candidate in a domestication locus, is highly related to seed size and oil content [42]. Moreover, luciferase complementation assays suggest an interaction between GmPDAT and GmDGAT1A [42]. The expression of *GmPDAT* increased the accumulation of PUFAs and total FAs, in parallel with the 100-seed weight, seed length and width in soybean [42]. Apart from enzymes of the Kennedy pathway, the production of PC-derived DAG, partially relying on phospholipase D (PLD) and PAHs (PA hydrolases) activities, may also contribute to oil accumulation. TAG content of *Camelina sativa* seeds was raised by 3% in lines co-expressing two isoforms of *AtPLDζ*s [137]. In soybean, 18 *PLD* members were further classified into six evolutionary branches, and work in *Arabidopsis* has demonstrated that the over-expression of *GmPLDγ* enhances seed TAG accumulation with favorable agronomic traits [43,117]. Despite its appeal as a target for lipid modification, it is noteworthy that the over-expression of *PLD* negatively affects seed vigor during natural ageing [118].

### 3.4. Enhancing Oil Accumulation by Blocking Lipid Hydrolysis

Fatty acids released from TAG (e.g., during germination) must be activated by long-chain acyl-CoA synthetase (LACS) to form acyl-CoAs for peroxisomal-mediated β-oxidation. The study of LACS subcellular localization, substrate specificity and tissue-specific expression patterns has enhanced our understanding of their function in oil flux [138]. While the functions of *AtLACS*s have been investigated in *Arabidopsis,* our understanding of their roles in common crops is very limited [139]. The contribution of LACSs to oil accumulation is dependent on their subcellular localizations [25,140]. GmACSL2, located in the peroxisome, is evolutionally different from other LACS-family members [25]. The over-expression of *GmACSL2* in yeast and soybean hairy root severely reduced lipid content [25]. Although this study may suggest that the suppression of *GmACSL2* could be a promising strategy for improving oil yields in the future, other studies indicated its potential negative impact on soybean growth. Its ortholog, *GmLACS2-3*, has been shown to be vital for cutin and suberin biosynthesis, and thereby important for abiotic stress tolerance [38].

During seed germination and seed maturation, TAG stored in lipid droplets is hydrolyzed to release free fatty acids (FFAs), followed by their transportation by an ABC transporter protein to peroxisomes where the FAs are further oxidized [22]. Blocking the degradation of TAG may prevent oil loss [141]. Candidate genes, such as triacylglycerol lipase SUGAR-DEPENDENT1 (*SDP1*), *SDP1-like* and Gly-Asp-Ser-Leu (GDSL)-motif lipases (*GDSL*s), have been shown to mediate seed lipid mobilization in *Arabidopsis* and rape [141,142,143]. Although *sdp1-5 sdp1L-2* double mutants show growth retardation on 1/2 MS medium, >90% of seed radicles emerged within 4 days, indicating that while TAGs are important storage compounds, they are not essential for germination or seedling establishment [141]. Knocking down all four *GmSDP1*s significantly increased the lipid content compared with that in the wild-type and also increased the ratio of oleic acid to linoleic acid in TAGs [46]. Disruption of *SDP1* exerted a positive impact on both seed yield and protein content, despite reductions of 15% in total raffinose and of 20–40% in the germination rate [15,46]. The selection of appropriate promoters (e.g., 2S albumin and 11S globulin) for silencing these lipases may retain desirable agronomic traits while producing more oil with UFAs or common TAGs [46,144]. Nevertheless, more detailed information is required regarding oil turnover for the biosynthesis of other biomolecules during seed desiccation in different species, as this may critically impact stress acclimation during post-germinative seedling development. A more precise characterization of this metabolic readjustment may be helpful in avoiding potential impacts on other agronomics during seed maturation. The disruption of *GmSDP1* in combination with the over-expression of an ATP-binding cassette (ABC) transporter gene (*GmABCA7*) may be an improved approach to minimize the detrimental effects on seed germination presented by the former alone [22].

GDSL lipases have also received considerable attention for seed oil improvement. The disruption of *BnGDSL1* promoted more oil accumulation ranging from 3.8–4.6%, but with an obvious germination penalty [143]. *BnGDSL1* mutations resulted in more oleic acid, similar to that with the RNAi-mediated silencing of all *GmSDP1*s in soybean [46,143]. In contrast, the over-expression of *BnGDSL1* or *AtGDSL1* in rapeseed greatly facilitated seedling establishment by substantially elevating the expression of β-oxidation-related genes, such as *KAT2*, resulting in transgenic lines producing about 5% less total oil [143]. Both *AtGDSL1* and *BnGDSL1* show strongest expression at the globular embryo stage of seed development in *Arabidopsis*. It is speculated that AtGDSL1 and BnGDSL1 may act as lipases when TAGs start to accumulate, but there is no experimental evidence to suggest that GDSL hydrolyzes TAG or other glycerolipids (e.g., DAG or MAG) in vitro. Some GDSLs are named as SEED FATTY ACID REDUCERs (SFARs) [145]. Two members of this *GDSL* family, *BnSFAR4* and *BnSFAR5*, have been associated with seed oil content in a whole-genome survey of all 111 *BnGDSL*s [145]. *SFAR* double mutants showed increased oil content of 8.7–12.1% with larger lipid droplets compared with those in its individual parent mutants [145]. Furthermore, the double mutation mitigated negative impacts under natural or osmotically induced stress environments [145]. Moreover, germination and seedling vigor in *SFAR* double mutants can be comparable to those of RS306, which may suggest that the functions of TAG hydrolysis, as mediated by SFAR, can be confined to seed development, rather than seed germination. However, 194 total *GDSLs* have been identified in the soybean genome [146]; it will be a formidable task to identify those members playing a significant role in oil improvement.

### 3.5. Enhancing Seed Oil Content through Manipulation of Transcription Factors

TAG biosynthesis is controlled by a complex regulation network of transcriptional factors (TFs). The identification of WRINKLED1 (WRI1) was a milestone discovery for our understanding of the regulation of TAG biosynthesis. Two *WRI1*s, *GmWRI1a* and *GmWRI1b*, have been identified in the soybean genome and *GmWRI1b* shows low expression in most soybean populations due to an alteration in its promoter region [147]. Both *WRI1*s can restore the *wri1* phenotypes defective in TAG accumulation, and they provide a new means to improve soybean oil content without impacting protein content [125,148]. A recent study has revealed the roles of *GmWRI1c* in oil accumulation and nodulation. The over-expression of *GmWRI1c* increased nodule numbers, which is probably attributed to a higher 16:0 composition and up-regulated nodulation genes like *GmNIN*. Furthermore, Haplotype analysis suggested that oil accumulation and nodule numbers of soybean cultivars may be positively associated with variations in the region of the *GmWRI1c* promoter during domestication [126]. However, some studies indicate that WRI1 may have profound impacts on plant growth other than oil accumulation. The over-expression of *GmWRI1b* in soybean activates *GmCYP714A*, which is associated with the deactivation of gibberellic acids (GAs), allowing for remodeling of the plant architecture and improved yields under field conditions [125]. Similarly, *AtWRI1* plays a role in mediating auxin homeostasis by altering the expression of genes related to auxin sensitivity and transport [149]. Unlike *WRI1b*, changes of some agronomic parameters, such as plant height and yields, have been avoided in *WRI1a*-over-expression lines under field experiments, although phenotypes were not stable [148]. *WRI1*s also regulate the carbohydrate distribution during nodule development and bidirectional symbiotic nutrient exchange in soybean [127]. Recently, GWAS analysis identified that *GmWRI14*, a *WRINKLED1-*like gene, is associated with the seed oil content and linoleic acid content [150]. The over-expression of *GmWRI14* in soybean increased oil content and 18:1 proportions through the down-regulation of *GmFAD2-1* and *GmFAD2-2b* [150]. Unexpectedly, the most pronounced effects of the heterologous expression *AtWRI1* in soybean were higher levels of palmitate (20%) rather than an increase in total oil content due to the up-regulation of *FATB* [151]. However, co-expressing *AtWRI1* and *AtDGAT1* (a push and pull strategy) failed to enhance the total FA content [152]. Stacking *AtWRI1* with *AtDGAT1* revealed an increase in starch production coupled with a roughly 15~25% decrease in yields, along with germination retardation [152]. The double transgenic plant reached a limit on the improvement of total FA content with a concomitant reduction in biomass production, which might be due to the low abundance of the lipid-droplet-packaging proteins oleosin and caleosin [152]. Collectively, plant hormone homeostasis, selection of the promoter and transgenic hosts should be considered when *WRI1* manipulation is applied to soybean oil improvement.

Additionally, LEAFY COTYLEDON2 (LEC2) is a positive regulator involved in lipid and protein deposition in *Arabidopsis* seeds. However, the sole *GmLEC2* in soybean was identified as a pseudogene and its syntelog relationship with different *LEC2s* among other legumes, such as common bean, has not been established [147]. Although two *GmLEC2*s with 97% similarity were identified in an updated version of the soybean genome (*G. max* Wm82.a4.v1), neither of them is expressed in all tissues (https://www.soybase.org; accessed on 8 March 2022). Intriguingly, *GmLEC2* is likely functionally replaced by *GmABI3b*, which plays *LEC2*-like roles in an *Arabidopsis lec2* mutant. A positive correlation was found between the expression levels of *GmABI3b* and seed oil content, identifying a potential target for further lipid content improvement [147].

## 4. Summary and Perspectives

Soybean is a major source of plant oil and protein for human and livestock consumption. The low seed oil content (20%) of native soybean requires improvement, while its fatty acid composition cannot meet some nutritional or industrial demands. Considerable effort has been expended in bioengineering soybean oil, and significant progress has been made in these genetic modifications, such as improving soybean oil enriched in oleic acid, linolenic acid, saturated FAs or unusual FAs. Because the biosynthesis of neutral glycerides shares upstream pathways with that of phospholipids, the modification of storage oil might interfere with or affect compositions of membrane lipids, which then may impact soybean growth and development as discussed in this treatise. Additionally, the protein content of soybean must be considered when oil content is improved, as the protein is highly valued as a co-product. Here, we have summarized some considerations or suggestions for soybean oil bioengineering to minimize their possible negative impacts on agronomic traits:

(1) As a paleopolyploid plant, soybean often has multiple paralogs, which presents difficulties for genetic modification approaches, such as the use of gene editing. On the other hand, functional differentiation of these paralogs presents opportunities for knocking out the main contributors to seed oil traits without impairing their paralogs in other tissues, such as *FAD2-1A/B* and *GmLOX1*, for the production of 18:1 and a beany flavor, respectively. (2) Seed-specific promoters may be applied for gene over-expression or silencing without seed-specific differentiation. For example, the disruption of *SDP1* or *SDP1-like* should be restricted to seed desiccation stages (or seed maturation), which will avoid germination retardation after imbibition. (3) Orthologs with high activity may be better alternative targets for modification over an endogenous gene with low activity, such as *GmFAD3*s. (4) The lipid metabolism of the host plays a critical role in maintaining healthy nodules and nodule-expressed lipid genes, such as *GmSACPD-C*, which should be carefully evaluated before the manipulation of FA composition. (5) Special care should be taken to manipulate acyltransferases with preferences for unusual FAs allowing their effective transfer from membrane lipids to neutral lipids. (6) Disturbances in storage protein biosynthesis may not be an advisable strategy for re-directing carbon flux towards oil accumulation. (7) Exploring beneficial genes or mutations via artificial selection, such as *GmST1* and *GmDGAT1A*, and mutations in *POWR1* and *B1* may expand the scope for oil enhancement with minor pleiotropic effects on soybean agronomic traits. (8) Soybean oil can be improved by over-expressing biosynthesis genes or positive regulators of TAG biosynthesis, such as *DGAT* or *WRI1*, respectively, and oil accumulation can be enhanced by blocking lipid hydrolysis by silencing *GmSDP1* or some *GmGDSL*s.

The development of co-ordinated multiple “omics”, highly efficient methods of gene-editing and pangenomic sequencing will further reveal the detailed mechanisms of TAG biosynthesis and its regulation. The functions of soybean lipid paralogs, not only in oil accumulation but also in plant growth and development and stress responses, will be elucidated, which will largely speed up genetic modification for oil improvement and enhance the predictability of their positive and negative impacts on soybean agricultural traits.

## Figures and Tables

**Figure 1 ijms-24-02256-f001:**
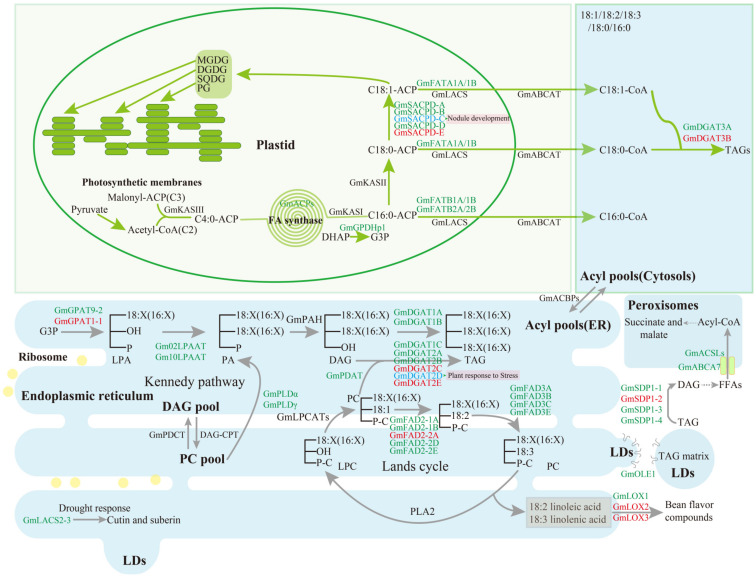
An overview of enzymes involved in FA biosynthesis, triacylglycerol assembly and degradation in soybean seeds. The yellow dots and green squares, in the figure, represent the ribosome and transmembrane proteins (ATP-binding cassette (ABC) transporter, GmABCA7), respectively. Abbreviations for substrates: DAG, diacylglycerol; DGDG, digalactosyldiacylglycerol; DHAP, dihydroxyacetonephosphate; FFAs, free fatty acids; G3P, glycerol-3-phosphate; LPA, *lyso*phosphatidic acid; LPC, *lyso*phosphatidylcholine; MGDG, monogalactosyldiacylglycerol; PA, phosphatidic acid; PC, phosphatidylcholine; PG, phosphatidylglycerol; SQDG, sulfoquinovosyldiacylglycerol; TAG, triacylglycerol. Abbreviation for proteins: ABCA or ABCAT, ATP-binding cassette (ABC) A transporter [22]; ACBPs, acyl-CoA-binding proteins [23]; ACP, acyl carrier protein [24]; ACSLs, long-chain acyl-CoA synthetase [25]; CPT, CDP-choline:diacylglycerol cholinephosphotransferase; DGAT, acyl-CoA:diacylglycerol acyltransferase [26,27,28,29]; FAD2, omega-6-desaturase 2 [30,31]; FAD3, omega-3-desaturase 3 [32,33]; FATA, acyl-ACP thioesterase A [34]; FATB, acyl-ACP thioesterase B [34,35]; GPAT, glycerol-3-phosphate acyltransferase [36]; GPDHp1, glycerol-3-phosphate dehydrogenase 1 [37]; KAS, 3-ketoacyl-[acyl carrier protein] synthase; LACs, long-chain acyl-CoA synthetase [38]; LOX, lipoxygenase [39]; LPAAT, *lyso*phosphatidic acid acyltransferase [40]; LPCAT, *lyso*phosphatidylcholine acyltransferase; OLE1, oleosin 1 [41]; PAP (or PAH), phosphatidic acid phosphatase; PDAT, phospholipid:diacylglycerol acyltransferase [42]; PDCT, phosphatidylcholine:diacylglycerol cholinephosphotransferase; PLA, phospholipase A; PLD, phospholipase D [43]; SACPD, stearoyl-ACP desaturase [44,45]; SDP1, SUGAR-DEPENDENT1 [15,46].

**Figure 2 ijms-24-02256-f002:**
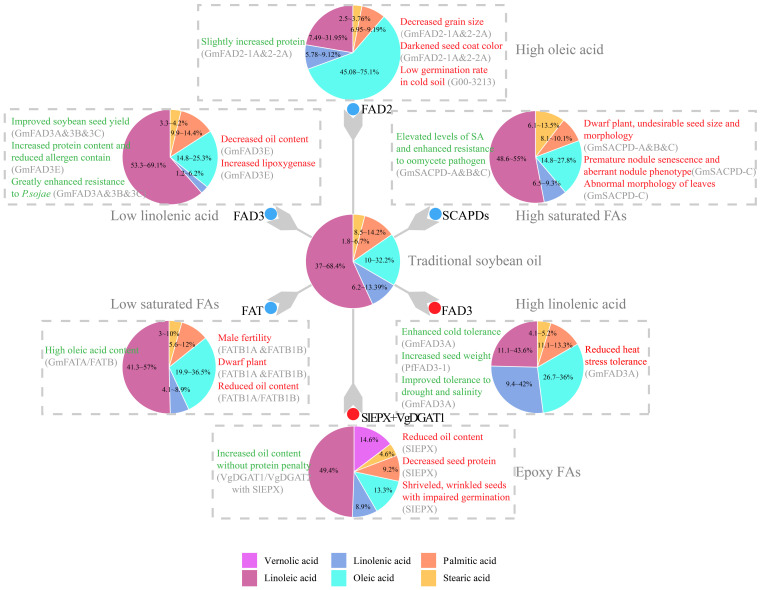
Modification of fatty acid biosynthesis pathways for desirable soybean oil improvements and the resultant impact on plant agronomics. Values in the pie chart designate the ranges of fatty acid composition in previous studies, while the area of pie slices illustrate their average value. For epoxy FAs, the line accumulating maximum level of vernolic acid were used for representation. As to genetic modification, genes with red dots are overexpressed, while genes with blue dots are knocked-out or knocked-down. Undesired or harmful effects (Cons) on soybean production are listed to the right of a pie and highlighted in red, while favorable or beneficial impacts (Pros) are listed to the left of the corresponding pie and highlighted in green. Abbreviation for proteins: FAD2, omega-6-desaturase 2 [14,52,53,64]; FAD3, omega-3-desaturase 3 [8,32,33,83,84,85,88]; FAT, acyl-ACP thioesterase [34,35]; SACPD, stearoyl-ACP desaturase [45,93,101,105]; SlEPX, epoxygenase [18,106,107].

**Figure 3 ijms-24-02256-f003:**
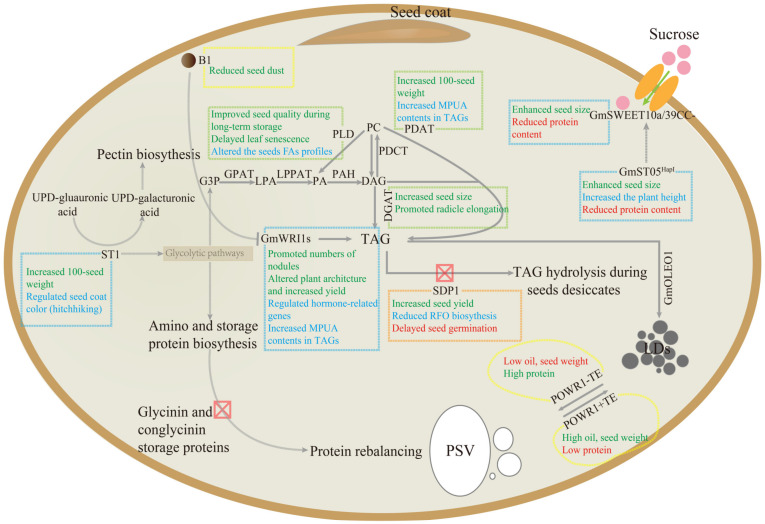
Schematic diagram of metabolic engineering for enhancing seed oil content (SOC) in soybean. Abbreviations: LDs, lipid droplets; PSV, protein storage vacuoles; B1, Bloom1 [116]; DGAT, diacylglycerol acyltransferase [26,28]; GPAT, glycerol-3-phosphate acyltransferase [36]; LPAAT, *lyso*phosphatidic acid acyltransferase [40]; OLE1, oleosin 1 [41]; PAH, phosphatidic acid phosphatase; PDAT, phospholipid:diacylglycerol acyltransferase [42]; PDCT, phosphatidylcholine:diacylglycerol choline phosphotransferase; PLD, phospholipase D [117,118]; POWR1, Protein Oil Weight Regulator 1 [119]; SDP1, Sugar-Dependent 1 [15,46]; ST05, Seed Thickness on Chromosome 5 [120]; ST1, Seed Thickness 1 [121]; SWEET10a, Sugar Transporter Gene 10a [122]; SWEET39, Sugar Transporter Gene 39 [123,124]; WRI1, WRINKLED1 [125,126,127].

## Data Availability

Not applicable.

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
