# Peer review of "Bioengineering of Soybean Oil and Its Impact on Agronomic Traits"

_ijms, 2023, doi:10.3390/ijms24032256_

Round 1
Reviewer 1 Report
The manuscript (ID:ijms-2150530) reviewed the progress on bioengineering of soybean oil and fatty acid composition. Moreover, the impacts on agronomic traits were also mentioned and discussed for these gene modifications. The manuscript is well organized and written, and comprehensive studies were described and discussed properly. In my opinion, the review part on fatty acid composition modification seems more comprehensive than that on bioengineering for soybean oil. That may be partly attributed to the limited studies for soybean oil modification. However, some essential progress on the key genes controlling soybean oil content is lacking in the manuscript. For instance, SWEET10a/b (the same gene as SWEET39) is assumed as a key gene coordinating yield and quality of soybean. The gene was only shown in Figure 3, but not mentioned in the manuscript. Another essential gene, GmST05, controlling seed size and oil & protein contents, should also be mentioned and discussed in this manuscript due to their potential application on oil improvement.
The detailed comments:
Page 6, line 249-261. The author mentioned “Oxidation of linolenic acid by lipoxygenases (LOXs) to produce conjugated unsaturated fatty acid hydroperoxides leads to an unpleasant beany flavor”. As we known, linoleic acid is also a polyunsaturated fatty acid, and the oxidation of linoleic acid could also lead to this beany flavor. Therefore, only reducing the level of linolenic acid cannot overcome the off-flavor problem.
Page 6, line 262-263. As abovementioned, in my opinion, the 18:2(linoleic acid) also positively correlates with beany flavor compounds. Therefore, only reducing FAD3 cannot solve this problem.
Page 6, line 271. What’s the full name of BADC? Please clarify it.
Page 11, Figure 3. GmSWEET39 and GmSWEET10a/b are same gene (Glyma.15G049200) with different names. Please revise this figure.
Reviewer 2 Report
This is a quite interesting review. Authors present up-to-date information about genetic modifications of soybean in the context of soybean oil modification. Some mistakes should be corrected.
General remarks
There are different font sizes in this text
If you use abbreviations you should explain them when the first time used, e.g. ER, ACP
Figures
I believe it is necessary to indicate sources on the basis of which figure 1 and 2 has been prepared.
I don't think underlining in Figures captions is necessary. Figure 2 - it is necessary to change the font color in the pie chart to black because now fig. 2 is barely readable.
Line 246 - something went wrong in this line
Round 2
Reviewer 1 Report
Comments:
A pronounced improvement has been achieved in the revised manuscript (ID:ijms-2150530). However, some minor mistakes are still found.
For instance, line 616-618, "B1, WRI1, ABI3, and LEC1" stand for proteins, and thereby should not be written in italic
Author Response
Comments
Comment 1: A pronounced improvement has been achieved in the revised manuscript (ID:ijms-2150530).
Response 1: Thank you for your positive comment.
Comment 2: However, some minor mistakes are still found.
For instance, line 616-618, "B1, WRI1, ABI3, and LEC1" stand for proteins, and thereby should not be written in italic.
Response 2: Thanks for your careful review. We have checked all the names of genes or proteins and have corrected all the errors. Others errors, such as an omission of “,” after e.g. or space between G. and max, have been revised. Additionally, we have added DOI numbers to all references.